# Governance quality indicators for organ procurement policies

**David Rodríguez-Arias**[1,2]*, **Alberto Molina-Pérez**[1,2,3], **Ivar R. Hannikainen**[1], **Janet Delgado**[2,4], **Benjamin Söchtig**[5], **Sabine Wöhlke**[2,5,6‡], **Silke Schicktanz**[5‡]

**1** FiloLab-UGR, Philosophy I Department, Universidad de Granada, Granada, Spain, **2** ELPAT-ESOT Public Issues Working Group, Santa Cruz de Tenerife, Spain, **3** Institute for Advanced Social Studies, Spanish National Research Council (IESA-CSIC), Córdoba, Spain, **4** University Hospital of the Canary Islands, Tenerife, Spain, **5** Department of Medical Ethics and History of Medicine, University Medical Center Goettingen, Goettingen, Germany, **6** Department of Health Sciences, HAW-Hamburg, Hamburg, Germany

‡ These authors are joint senior authors on this work.

* dra@ugr.es

**Data Availability Statement:** Data are available in OSF: https://osf.io/75gqh/.

**Funding:** DRA MINECO FFI2017-88913-P MINISTERIO DE ECONOMÍA Y COMPETITIVIDAD https://administracion.gob.es/pagFront/

## Abstract

### Background

Consent policies for *post-mortem* organ procurement (OP) vary throughout Europe, and yet no studies have empirically evaluated the ethical implications of contrasting consent models. To fill this gap, we introduce a novel indicator of governance quality based on the ideal of *informed support*, and examine national differences on this measure through a quantitative survey of OP policy informedness and preferences in seven European countries.

### Methods

Between 2017–2019, we conducted a convenience sample survey of students ($n = 2006$) in Austria (AT), Belgium (BE), Denmark (DK), Germany (DE), Greece (GR), Slovenia (SI) and Spain (ES), asking participants about their donation preferences, as well as their beliefs and views about the policy in place. From these measures, we computed indices of informedness, policy support, and fulfilment of unexpressed preferences, which we compared across countries and consent systems.

### Results

Our study introduces a tool for analyzing policy governance in the context of OP. Wide variation in policy awareness was observed: Most respondents in DK, DE, AT and BE correctly identified the policy in place, while those in SI, GR and ES did not. Respondents in opt-out countries (AT, BE, ES and GR) tended to support the policy in place (with one exception, i.e., SI), whereas those in opt-in countries (DE and DK) overwhelmingly opposed it. These results reveal stark differences in governance quality across countries and consent policies: We found a preponderance of informed opposition in opt-in countries and a general tendency towards support–either informed or uninformed–in opt-out countries. We also found informed divergence in opt-in countries and a tendency for convergence–either informed or uninformed–among opt-out countries.

espanaAdmon/directorioOrganigramas/ficha
UnidadOrganica.htm?idUnidOrganica=89063&
origenUO=gobiernoEstado&volver=gobierno
Estado#.X_7hO5NKhBw The funders had no role in
study design, data collection and analysis, decision
to publish, or preparation of the manuscript.

**Competing interests:** The authors have declared
that no competing interests exist.

## Conclusion

Our study offers a novel tool for analyzing governance quality and illustrates, in the context
of OP, how the strengths and weaknesses of different policy implementations can be esti-
mated and compared using quantitative survey data.

## Introduction

Policies for deceased organ procurement (OP) (i.e., laws, official guidelines) aim to increase
transplantable organs supply while honoring societal values regarding bodily self-determina-
tion and the afterlife. International rankings of countries' success regarding OP focus on will-
ingness to donate and donation activity [1–4], while paying less attention to policies' capacity
to encompass societal values. In this article, we propose and test–through a cross-European
survey–a novel framework of analysis that focuses on the latter.

Consent policies for *post-mortem* OP establish how citizens may express their preferences
(e.g., donor and refusal registries, donor cards, etc.), next-of-kin's role in OP decision-making,
and the default rule (retrieving or not retrieving the organs) when no preference is available.
Jurisdictions vary regarding the means of expressing individuals' preferences and the authority
of the family [5]. In Europe, most countries have implemented an opt-out (*presumed consent*)
model, while some countries retain the opt-in (*explicit consent*) model. The difference between
these regulations lies in the *default* option that applies when the deceased person has not
expressed their preference. In opt-out systems the organs may be retrieved, whereas in opt-in
systems they may not.

Proponents of opt-out policies claim that, because organs are procured from dead individu-
als, this policy cannot significantly harm donors' interests. Critics suggest, however, that pre-
sumed consent may undermine individual autonomy–namely, by overriding the posthumous
interests of those who did not want to donate and yet failed to express their preference [6, 7].
The World Health Organization recognizes that this is particularly problematic for those who
are unaware of their default status as donors, thus requiring "that people are fully informed
about the policy and are provided with an easy means to opt out" [8]. Importantly, however,
opt-in policies are subject to similar objections. Opt-in may also violate the posthumous inter-
ests of individuals willing to donate who failed to express that preference, potentially due to
ignorance of the policy in place [9]. Hence, both systems may violate the autonomy and post-
humous wishes of those who–whether through unawareness or incapacity–do not take the
necessary steps to revert their default status (e.g., unregistered non-donors in opt-out coun-
tries, and unregistered donors in opt-in countries).

### Shared health governance on organ procurement policies

These issues related to both opt-in and opt-out raise the important moral and political ques-
tion of what constitutes optimal governance of OP policy. In democratic, liberal societies, gov-
ernance of health systems should be based on shared knowledge, sufficient deliberation, as
well as inclusive decision-making and shared responsibility. Jennifer Ruger has called such a
system *shared health governance* (SHG) [10], which requires the engagement and goal
endorsement of an informed public.

On the one hand, public involvement–from national surveys to public deliberative pro-
cesses–is advisable when policy-making deals with conflicting values about the public good or
potentially divisive topics whose elucidation requires a combination of technical and non-

expert like knowledge [11]. On the other hand, to be efficient and sustainable, SHG stresses the importance of both external motivation (e.g., government policy enforcement) and internal motivation (i.e., individual actors internalizing public moral norms so that their cooperative behaviors are autonomously motivated, with no need to appeal to social sanctions). For individuals to autonomously embrace their societal co-responsibilities and meaningfully engage in policy-making, they need to be properly informed of what is at stake [10]. However, ideal governance should also weigh and integrate competing values and goods in the interest of a pluralistic society. Where competing values and goods are at stake, governance would strive for policies that can be supported by a majority, while minimally hampering the values of opposing minorities. Pluralism regarding value-laden assumptions towards death, the body, and religion impacts people's attitudes towards *post-mortem* organ donation [12, 13]Optimal governance on these policies should therefore balance the aspiration to implement prevailing values by way of a specific policy (e.g., opt-in or opt-out), with the goal of accommodating the minorities whose values conflict with extant policy. Awareness of the public's attitudes regarding donation and transplantation can help policy-makers develop palatable and effective policies [14]. The way health policy-making recognizes the interests of *policy dissenters* and *policy opposers* can therefore be seen as a measure of its moral and democratic quality. In our terminology, policy *dissenters* are people whose *personal preferences* towards donation differ from the system's default or standard option (e.g., donors in opt-in countries, and non-donors in opt-out countries). Dissenters may or may not be aware of the policy actually in place in their country, and may not have expressed their preferences (e.g., unregistered donors in opt-in countries, and unregistered non-donors in opt-out countries). *Policy opposers* are those whose *policy preferences* differ from the policy in place (e.g., opt-out supporters living in opt-in countries, and opt-in supporters living in opt-out countries).

At the same time, OP policies need to address organ shortage in ways that minimally harm or wrong potential donors and their families [15]. Ideally, OP policies should make use of default rules in ways that simultaneously foster the general interest of organ transplantation and enable the expression of legitimate forms of disagreement and pluralism. For example, those who want to donate their organs in an opt-in model (where the default rule is not to donate) should have fair and accessible options to signal their willingness to donate, and their wish should be fulfilled. Correspondingly, those unwilling to donate in a presumed consent model should have fair and accessible chances to opt-out. According to the SHG model, *fair and accessible* options involve informed individuals who can make both meaningful judgments about their preferred policy, and autonomous decisions to have their posthumous interests fulfilled. Public knowledge of national regulations regarding automatic processes for OP affects how these policy trade-offs between individual and collective interest are made, and becomes a key element for assessing the governance quality of OP consent policies.

## Public knowledge in health governance

From a collective or political stance, knowledge is necessary for citizens' understanding and meaningful engagement in socially controversial health policy debates, as it enables them to exert their democratic right to dispute or meaningfully endorse the *status quo*.

From the individual or moral stance, knowledge is necessary for individuals to act so that their autonomous preferences (e.g., their posthumous wishes) are met. This is especially true for policy *dissenters*: people whose preferences regarding donation do not match the standard option set by their national policy.

In what follows, we first propose a theoretical framework for analyzing consent policies for deceased OP from the perspective of health governance. Then, we empirically test this

framework by assessing and comparing the governance quality of seven European countries–Austria, Belgium, Denmark, Germany, Greece, Slovenia, and Spain. We do so by administering a survey to a convenience sample of university students focusing on: a) participants' views on public involvement in policy-making regarding the consent system for OP, b) willingness to donate one's organs, c) expression of individual preferences regarding one's OP, d) attitudes toward consent models, and e) knowledge of the respective consent model. Our sample (N = 2006) serves the purpose of testing the theoretical framework, but the respondent population is highly selective and cannot be considered representative of national populations, as discussed below.

## Health governance quality assessment: A theoretical framework

Our analysis is divided in two dimensions that correspond to the two aforementioned indicators of optimal governance:

  I. the beneficent capacity to embody predominant values through consent policy, and

  II. the non-maleficent capacity to observe (i.e., minimize violations of) the autonomy and posthumous interests of policy-dissenting individuals.

*Indicator I* assesses the policy goal of *representing* the values of the majority, whereas *Indicator II* assesses the policy goal of *preserving* the values of policy-dissenting individuals. For each dimension, we identify four levels of governance quality, ranked from best (*A*) to worst (*C*). Two levels that neither correspond to best-case nor worst-case scenarios but are suboptimal in different ways (degree of policy awareness, and degree of policy agreement) are labelled as $B_1$ and $B_2$.

### Indicator I: Consent policies' capacity to represent the values of the majority

*Indicator I* reflects variation in public *knowledge* and *support* of the consent policy in place. The combination of these two measures creates four governance levels, which are applicable to both opt-in and opt-out countries (Fig 1):

*Level A* (*Informed support*) takes place when the public knows and supports the policy in place in their country. The majority of the population is aware of the consent policy in place (either opt-in or opt-out), and agrees with that policy. This is arguably the best-case scenario in terms of health governance. Not only does it foster deliberative public engagement but it also decreases the likelihood of disrespecting citizens' shared values.

*Level $B_1$* (*Informed opposition)* occurs when the public knows the policy in place but only a minority supports it. This scenario reflects a majority enlightened opposition to the actual policies. The public is empowered to criticize the *status quo* and can exert their political right to demand political alternatives. *Informed opposition* may lead to policy reform, as the opposing majority may democratically push to replace the extant policy. However, in terms of deliberation quality, this situation is satisfactory, as society is aware of the mismatch between its predominant preferences and the enacted policy, which reduces the perception of policy manipulation or abuse.

*Level $B_2$* (*Uninformed support*) occurs when people falsely believe that the policy in place is different from the one they support (i.e., they are unaware that the policy they prefer or support is in fact in place in their own country). This scenario may have different consequences depending on the consent model in place. If the majority prefers opt-out and mistakenly believes that opt-in is in place, they might unnecessarily take extra efforts to state their

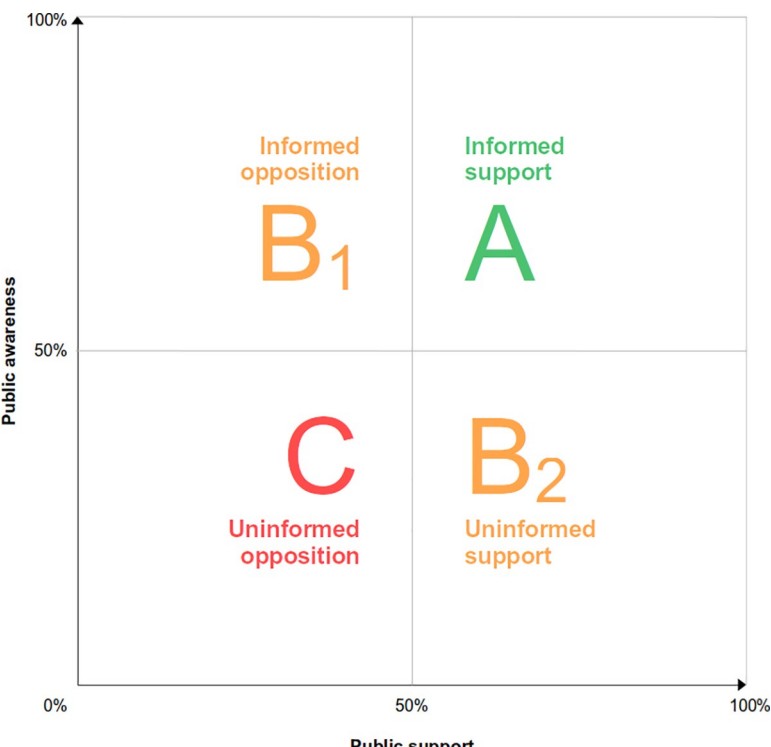

**Fig 1. Relation of level of knowledge and supportive attitudes of the public towards consent policies for organ procurement.**

willingness to donate. Conversely, if the majority prefers opt-in and mistakenly believes that the enacted policy is opt-out, those refusing organ donation may unnecessarily register their refusal. Uninformed support does not lead the public to critically challenge the existing policy, as most do favor the model after all. However, it hinders deliberative public engagement, a problem that policy makers may address by better informing the public.

*Level C* (*Uninformed opposition*) appears when the public neither knows nor supports the policy in place. In opt-out countries, this would occur if most people reject presumed consent policies *and* ignore that presumed consent is the enacted policy or, in opt-in countries, if most people reject the explicit consent policy *and* ignore that explicit consent is in place. Uninformed opposition is arguably the worst scenario from a health governance perspective. As the public ignores that they are under a system they furthermore disagree with, they are thus deprived from the possibility of critically engaging with the policy, resulting in lower deliberative capacity. In terms of policy stability, lack of awareness likely results in fragile endorsement of that policy in the short term, and the risk of perceived deceit, which may potentially result in public hostility against the policy in the long run.

## Indicator II. Consent policies' capacity to avoid violations of policy-dissenters' unexpressed preferences

Mechanisms for declaring personal preferences regarding *post-mortem* OP–e.g., donation and refusal registries, official donor status cards, or advance directives–can be more or less conducive to the fulfilment of posthumous interests and autonomy. Having easy access to such procedures is important for individuals to have their preferences fulfilled, especially for dissenters,

who won't automatically have their preferences fulfilled. Despite the availability of registries, people often do not record their actual preferences regarding OP [16].

Both opt-in and opt-out assume a risk of error in their respective presumptions about those who fail to express their preference. The degree of that risk can be assessed by ascertaining the proportion of people who simultaneously a) have a preference regarding organ removal that is contrary to the default course of action, and b) have not expressed that preference through binding means (i.e., unregistered dissenters). To assess the extent to which a policy will likely contravene the unexpressed preferences of the dissenters, a proper health governance assessment of consent policies for deceased OP needs to consider the *prevalence of unexpressed preferences* (the number of *silent individuals*), and *whether such preferences are convergent* or *divergent with the default option* established by the policy.

- *Preference-policy convergence* occurs either when the deceased wished to be a donor and the standard is to remove organs (opt-out), or when the deceased did not wish to be a donor and the standard is not to remove organs (opt-in). Under any of these circumstances, the unexpressed preferences are likely to be fulfilled.

- *Preference-policy divergence* exists when the deceased either did not wish to donate in opt-out or wished to donate in opt-in. Under these circumstances, the deceased's unexpressed preferences are unlikely to be fulfilled. Consent policies should aim at *minimizing divergence* between the deceased's non-expressed preferences and the policy's presumption.

Importantly, failure to express a preference does not necessarily mean that the person had no preference whatsoever. People may fail to express their preferences for different reasons, including lack of awareness of the necessity to do so, lack of knowledge about the available procedures to do so, lack of capacity, and personal choice.

Individuals can exercise their autonomy about organ donation by expressing their posthumous preferences, or by choosing to remain silent. In the latter case, their decision can be deemed autonomous only if they realize the consequences of failing to express a preference, given the policy in place. On the contrary, ignorance of the policy should be taken as an indicator of defective health governance if it accounts for people not expressing their preferences *and* such preferences differ with the standard course of action established by default rule of the consent model in place. In order to evaluate the extent to which policy unawareness (as opposed to personal choice) engenders violations of dissenters' preferences, awareness of the policy in place is included as a third variable in this health governance quality indicator. By combining convergence/divergence of non-expressed preferences with awareness of the policy, we can again distinguish four levels of governance quality (valid for both opt-in and opt-out countries): *A', B₁', B₂'* and *C'* (Fig 2):

*Level A'* (*Informed convergence*) combines high knowledge of the policy (as a condition for autonomous decision-making) and prevalent convergence between the policy's presumption and the deceased's actual wishes (as a condition for the fulfilment of non-expressed preferences).

*Level B₁'* (*Informed divergence*) combines high knowledge of the policy with low convergence rates, which implies that non-expressed preferences remain unfulfilled. This is the case, for example, when people in opt-in countries, despite wanting to be organ donors and despite knowing that they need to register their decision, fail to do so. Level B₁ is suboptimal as it might harm dissenters' interests in donating or keeping their organs (in opt-out dissenters), but that harm cannot be attributed to the policy. As dissenters knew about the implications of staying silent, this situation can also be interpreted as an expression of indifference, apathy, ambivalence, or moral uncertainty.

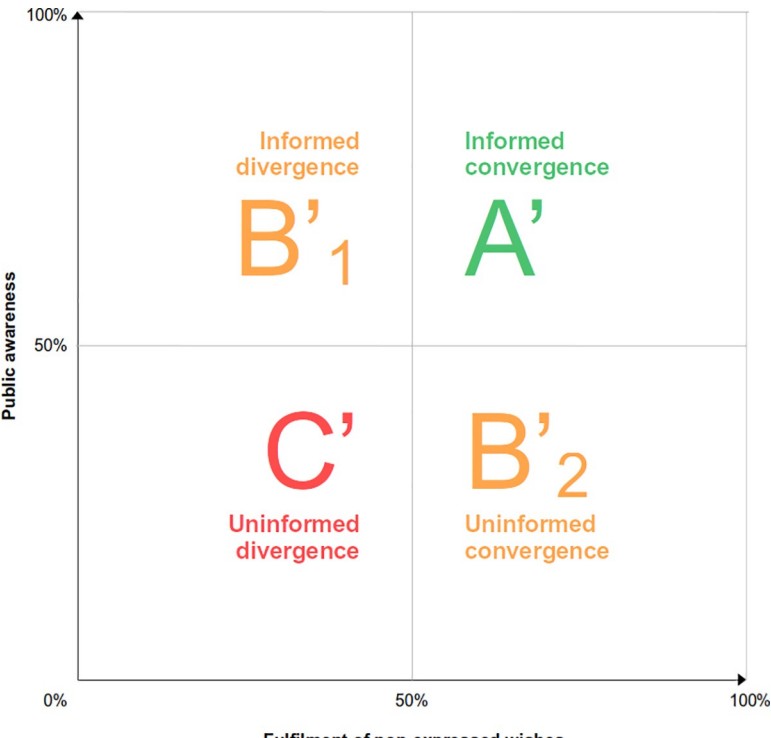

**Fig 2. Relation of level of knowledge and the fulfilment of non-expressed preferences for organ procurement.**

*Level $B_2$'* (*Uninformed convergence*) combines low knowledge of the policy with high convergence rates. It occurs when people are unaware of the policy in place, but their non-expressed wishes are nevertheless fulfilled (accidentally). This is the case, for example, when donors in opt-out countries fail to register their decision to donate, despite their erroneous belief that the policy in place is opt-in—a situation that may be quite common [17].

*Level C'* (*Uninformed divergence*) combines low knowledge of the policy with high divergence rates. It occurs when people are unaware of the policy in place and have their non-expressed wishes unfulfilled as a result of the application of the default. In opt-out jurisdictions this would be the case when non-donors fail to express their refusal on the wrong assumption that the policy in place is opt-in. In opt-in jurisdictions, Level C' occurs when donors fail to record their consent spurred by the erroneous belief that opt-out is applicable. Under Level C' policy-dissenters' preferences are not acknowledged by the policy. Besides, it hinders public deliberation. In this respect, Level C' is arguably the worst scenario in terms of *Indicator II* for health governance quality assessment.

## Methods

### Recruitment methods and locations

Between October 2018 and November 2019, participants were recruited in seven different countries by using two convenience sampling methods: An online survey link distributed via university mailing lists, and flyers displayed on campus at the following universities: Medical University Innsbruck and Private University for Health Science, Medical Informatics and Technology (Austria), University of Antwerp (Belgium), University of Copenhagen (Denmark), University of Göttingen (Germany), "Alexander" Technological Educational Institute

**Table 1. Sample size, median age and gender distribution by country.**

| Policy in place | Country | Code | N (participants) | Age Bracket (Median) | Gender (% women) |
|---|---|---|---|---|---|
| Opt-in | Denmark | DK | 230 | 20 to 24 | 72% |
| | Germany | DE | 424 | 20 to 24 | 75% |
| Opt-out | Austria | AT | 339 | 25 to 29 | 74% |
| | Belgium | BE | 439 | 20 to 24 | 76% |
| | Greece | GR | 159 | 20 to 24 | 78% |
| | Slovenia | SI | 190 | 20 to 24 | 72% |
| | Spain | ES | 222 | 20 to 24 | 73% |

(Greece), University of Ljubljana (Slovenia), and University of Granada (Spain). Student participants in Germany, Greece, Denmark, Slovenia, and Spain were offered participation in a 50€ e-commerce voucher lottery. At each participating site, partners were asked to recruit at least 100 students of Health Sciences/Medicine and 100 students of Social Sciences/ Humanities.

## Sample composition

Overall, 2006 Austrian, Belgian, Danish, German, Greek, Slovenian and Spanish students (age bracket (mode) = 20 to 24 years) took part in our study. A majority of them (74%) were women (see Table 1), which partially reflects an over-representation of women in the disciplines concerned [S1 File] [18] but may also account for an answer bias, as mentioned in the Discussion section. Approximately half of the sample were health sciences and medicine students, and the remaining half were students of a broad range of humanities and social science disciplines (see Table 2).

## Study design and online survey

We conducted an online survey using LimeSurvey© software. An initial group of partners started in autumn of 2018. The second round started in the spring of 2019. Data collection differed due to differences in university term schedules. The questionnaire was designed by the

**Table 2. Frequency table of field of study.**

| Field of study | | n (participants) |
|---|---|---|
| Non-health related studies | Anthropology | 51 |
| | Economics | 183 |
| | Humanities | 288 |
| | Philosophy | 52 |
| | Social Science | 370 |
| | Social Work | 33 |
| | Sociology | 21 |
| | Total | 998 |
| Health related studies | Health Science | 237 |
| | Medicine | 497 |
| | Nursing | 218 |
| | Public Health | 56 |
| | Total | 1008 |
| **Total** | | **2006** |

members of a working group (https://esot.org/elpat/structure/) and agreed upon in its English version. Prior to data collection, the questionnaire was translated into the primary official language for each participating country and back-translated for quality control [19].

All procedures performed in this study were carried out in accordance with the European Charter of Fundamental Rights and with the Declaration of Helsinki and its later amendments. The authors have obtained the required permits and approvals and have ensured that the study complies with local ethical and legal requirements for all countries involved. The study protocol was reviewed and approved as minimal risk research by the University Medical Center Göttingen Human Research Review Committee (Ref. no.: 13/01/19), as well as by the Universidad de Granada Ethics Committee on Human Research (ref. no. 718/CEIH/2018). For the other countries involved (Belgium, Denmark, Greece, Slovenia, and Austria), no extra ethics review was required for this type of study. Written informed consent was obtained from all individual participants prior to participation. A blank example of the form used to inform participants is included in the S1 File.

The questionnaire (S1 File) features a total of 36 questions grouped into six categories, and asks about participants' a) prior experience, b) knowledge and c) personal views on organ donation, and donation policies, d) reasons for donating/not donating, body concepts, attitudes towards brain death and alternatives to posthumous transplantation, e) the public and policy discourse about organ donation. Additionally, participants are asked to report sociodemographic information including: their age (in multiple-age brackets), gender, religiosity, and field of study. The structure and content of the survey was based on a prior survey developed by members of the team [12, 20]. Participants were informed in advance about the nature, purpose and duration of the study, and asked to provide informed consent if they wished to participate. Further information about the project was provided at the end of the survey through a link to the project website (issatosurvey.wordpress.com).

## Analysis approach

Statistical analyses were conducted in *R* version 4.0.2., by employing the *stats*, *lmerTest* and *emmeans* packages. National comparisons of knowledge and expression status (dichotomous variables) were conducted using chi-squared tests of non-independence. Analyses involving interval variables (i.e., support, attitudes toward consent systems) employed one-sample and paired *t*-tests. Further multivariate analyses (e.g., to examine demographic differences in knowledge or attitudes) were logistic and linear regressions, respectively.

To evaluate whether our sample size afforded adequate statistical power to examine variation in knowledge and attitudes across countries, we conducted sensitivity power analyses for one-sample proportion tests (i.e., knowledge) and *t*-tests (i.e., attitudes). Setting the alpha-level to .05 and power (1—beta) to .80, a per-country target sample size of 200 (mean *n*/country = 287) enabled our study to reliably detect small effects on knowledge (Cohen's $g$ = .10) and attitudes (Cohen's $d$ = 0.20) in each country. In turn, our aggregate sample size ($N$ = 2006) provided almost perfect power (> 99%; alpha = .05) to detect small effects even in multivariate regression analyses (i.e., $f^2$ = .02). Study data have been made available on the *Open Science Framework* at https://osf.io/75gqh/.

## Results

We focus our analyses on data about participants' views on public involvement in policy-making regarding the consent system for OP, reported general knowledge about OP regulation, and the twofold governance quality assessment. For each assessment, we examine national variation in governance quality through the lens of our four-tier scheme.

**Table 3. Involvement of the public in policy-making and reported knowledge about organ donation.**

| Policy in place | Country | *"The public should be involved in discussions about legal changes on the consent system for organ donation.* Mean agreement on a 6-point Likert scale, from 1 = 'totally disagree' to 6 = 'fully agree', and standard deviation (in brackets). | *"Do you feel sufficiently informed about the topic of organ donation?"* Frequency and percentage (in brackets). | | |
|---|---|---|---|---|---|
| | | | Yes | No | Don't know |
| Opt-in | DE | 5.07 (1.21) | 188 (44) | 187 (44) | 49 (12) |
| | DK | 4.68 (1.40) | 108 (47) | 101 (44) | 21 (9) |
| Opt-out | AT | 4.45 (1.64) | 141 (42) | 165 (49) | 33 (10) |
| | BE | 4.10 (1.48) | 123 (28) | 287 (65) | 29 (7) |
| | ES | 4.82 (1.42) | 41 (18) | 164 (74) | 17 (8) |
| | GR | 5.03 (1.16) | 16 (10) | 136 (86) | 7 (4) |
| | SI | 4.61 (1.39) | 15 (8) | 159 (84) | 16 (8) |

## Views on public involvement in policy-making and self-reported knowledge about organ donation

Our first analysis step includes a test of participants' attitudes toward a basic premise of the SHC concept, namely, the preference for public involvement in policy decisions. To assess this question, we conducted a series of one sample $t$-tests against the point of neutrality ($\mu = 3.5$). In every country, respondents tended to consider that the public should be involved in deliberation regarding the consent system for organ donation, all $t$s > 8, all $p$s < .001. As shown in Table 3, German and Greek participants were among the most supportive of public engagement, while Belgian participants were among the least.

Turning to self-reports of informedness, we observed variation across countries, $\chi^2(df = 12) = 210.8$, $p < .001$. Generally speaking, a minority of students reported feeling sufficiently informed about organ donation, with percentages ranging from more than 40% in Denmark, Germany and Austria, and 28% in Belgium, to less than 20% in Spain, Greece, and Slovenia.

## Knowledge rates by country

To complement the subjective, self-reported assessment of information, we test for the objective level of knowledge (policy awareness). For each country, we define participants' *knowledge* status as whether they correctly reported the policy in place in their country of residence–from among a set of three response options ("Presumed Consent", "Informed Consent" and "Other") and a non-response option. In a chi-squared test, we found wide variation in knowledge rates across countries, $\chi^2(df = 6) = 933.0$, $p < .001$. A vast majority of Danish (97%) and German (92%) respondents correctly reported that their country has an opt-in policy. Similarly, most Austrian (88%) and Belgian (71%) respondents correctly answered that an opt-out policy is in place. Meanwhile, Spanish (35%), Greek (11%), and especially Slovenian (5%) participants remained largely unaware of the fact that their countries have opt-out policies (Table 3).

To further examine individual differences in knowledge status, we entered a series of socio-demographic predictors in a multiple, logistic regression. Awareness of the consent system in

place was higher among health sciences and medicine students than among humanities and social sciences students ($z = 4.61$, $p < .001$), even when controlling for gender, nationality and religiosity. Meanwhile, religiosity ($z = -1.33$, $p = .18$) and gender ($z = -0.67$, $p = .51$) did not independently predict knowledge status.

### Attitudes toward opt-in and opt-out systems

Participants were asked to report their attitudes toward each of four policy alternatives on separate 6-point Likert scales (anchored at 1 = "Fully Disagree" and 6 = "Fully Agree"): opt-out, opt-in, mandatory choice, and mandatory procurement (organ conscription). At the aggregate level (i.e., collapsing across countries), presumed consent ($M = 4.68$, $SD = 1.71$) emerged as the preferred consent system in paired $t$-tests against all other response options (all $ps < .001$). Mandatory choice ($M = 4.32$, $SD = 1.69$) also elicited favorable attitudes, whereas explicit consent ($M = 3.16$, $SD = 1.83$) and especially conscription ($M = 1.84$, $SD = 1.33$) elicited unfavorable attitudes overall.

We also examined differences among sociodemographic groups in two separate multiple linear regressions: one, predicting attitudes toward presumed consent; another predicting attitudes toward explicit consent. After controlling for national differences (dummy-coding the country variables), we observed significant effects of religiosity and health studies, but not gender, on attitudes toward opt-in and opt-out systems: Specifically, students of health-related disciplines reported more favorable views about presumed consent, $B = 0.28$, $t = 3.52$, and more unfavorable views about explicit consent, $B = -0.29$, $t = 3.52$, both $ps < .001$. Religiosity exerted opposite effects in both models, such that religious participants were more opposed to presumed consent, $B = -0.38$, $t = 7.93$, and more in favor of explicit consent, $B = 0.30$, $t = 6.12$, both $ps < .001$.

**Consent policy support rates by country.** Next, we sought to understand whether countries differ in the extent to which participants favor the policy in place *in their own country*. For instance, support (/opposition) among German students would be defined by their favorable (/unfavorable) attitudes toward the opt-in system, whereas support (/opposition) among Spanish students would be defined by their favorable (/unfavorable) attitudes toward the opt-out system. We then conducted a series of one-sample t-tests against the point of neutrality ($\mu = 3.5$), interpreting significant differences between national means and the scale midpoint as either a tendency to support the policy in place (if the national mean exceeds the midpoint), or a tendency to *oppose* the policy in place (if the national mean falls below the midpoint).

As shown in Table 4, respondents in Austria, Belgium, Spain, and Greece held favorable attitudes towards the system in place in their country (opt-out). Meanwhile, participants in Slovenia (opt-out) and Germany (opt-in) were ambivalent, and Danish students tended to hold negative attitudes toward opt-in. Notably, participants in opt-in countries revealed

**Table 4. Knowledge and support rates by consent policy and country.**

| Policy in place | Country | Knowledge of the policy in place (%) | Support of the policy in place | | |
|---|---|---|---|---|---|
| | | | % | Mean *(SD)* | *T*-test against point of neutrality (*mu* = 3.5) |
| Opt-in | DE | 92 | 41 | 3.38 *(1.71)* | $p = .13$ |
| | DK | 97 | 47 | 3.15 *(1.60)* | $p = .001$ |
| Opt-out | AT | 88 | 86 | 5.20 *(1.45)* | $p < .001$ |
| | BE | 71 | 73 | 4.98 *(1.47)* | $p < .001$ |
| | ES | 35 | 72 | 4.63 *(1.73)* | $p < .001$ |
| | GR | 11 | 60 | 4.06 *(1.86)* | $p < .001$ |
| | SI | 5 | 48 | 3.56 *(1.82)* | $p = .67$ |

**Table 5. Health governance indicator I: Policy capacity to represent the preferences of an informed majority.**

| Policy in place | Country | n | Scenarios | | | |
| --- | --- | --- | --- | --- | --- | --- |
| | | | A | $B_1$ | $B_2$ | C |
| | | | *Informed support* | *Informed opposition* | *Uninformed support* | *Uninformed opposition* |
| Opt-in | DE | 424 | 177 (42%) | 213 (50%) | 22 (5%) | 12 (3%) |
| | DK | 230 | 93 (40%) | 131 (57%) | 1 (0%) | 5 (2%) |
| Opt-out | AT | 339 | 260 (77%) | 38 (11%) | 30 (9%) | 11 (3%)) |
| | BE | 439 | 289 (66%) | 24 (5%) | 85 (19%) | 41 (9%) |
| | ES | 222 | 67 (30%) | 10 (5%) | 95 (43%) | 50 (23%) |
| | GR | 159 | 10 (6%) | 8 (5%) | 85 (53%) | 56 (35%) |
| | SI | 190 | 5 (3%) | 5 (3%) | 86 (45%) | 94 (49%) |

considerable opposition to the system in place–as foreshadowed by the previous analysis of attitudes.

## Assessment of organ procurement governance quality

**Indicator I. Consent policies' capacity to represent the preferences of the majority.** Having defined both knowledge and support for each participant, we now tabulate these variables to assess consent policies' capacity to encompass commonly shared values and to represent the preferences of the majority that should ideally be informed. The assessment of governance quality differs across countries and consent policies. (Table 5 and Fig A in S1 File) Substantial *informed opposition* (Level B1) arose in opt-in countries (Germany = 50.2%; Denmark = 57.0%), with at least half the sample falling into this quadrant (Table 5). In both countries, knowledge rates are high, which–when paired with generally unfavorable ratings of opt-in systems–result in high levels of *informed opposition*. In opt-out countries, the assessment of governance quality was more diverse. Several opt-out countries revealed a prevalence of *uninformed support* (Level B2) (Spain = 42.8%; Greece = 52.8%; Slovenia = 45.3%), and Slovenia, Greece and Spain revealed considerable levels of *uninformed opposition* (Level C) (Slovenia = 49%; Greece = 35%; Spain = 23%). Austria and Belgium revealed a predominance of *informed support* (Level A) (Austria = 76.7%; Belgium = 65.8%).

To facilitate inter-country comparison, in Fig 3 we display the prevalence of knowledge (participants who are informed about the policy in place) and support (participants who favor the policy in place) for each country. Austria and Belgium are situated overall in Level A of governance quality (*informed support*) because they combine higher public knowledge and higher support for the policy in place. In Denmark and Germany, public knowledge of the policy is very high (more than 90%) but fewer than 50% of respondents support the policy, thus situating these two countries in the suboptimal category Level $B_1$ (*informed opposition*). In Spain and Greece, a majority support the national policy (opt-out) though few know it is in place, thus situating these countries in suboptimal Level $B_2$ (*uninformed support*). Finally, Slovenia is situated in Level C category (uninformed opposition) since it combines low public knowledge with slightly less than 50% support for the policy (opt-out).

**Indicator II. Consent policies' capacity to preserve the values of dissenting individuals.** To assess policies' capacity to honor the posthumous preferences of dissenting individuals, we focus on (1) unexpressed preferences (by participants who remain silent regarding organ procurement) that (2) converge or diverge from the policy standard in each country (Table 6). Additionally, to assess whether or not the unexpressed preferences are autonomous, we tabulate participants' convergence/divergence against their awareness/unawareness of the

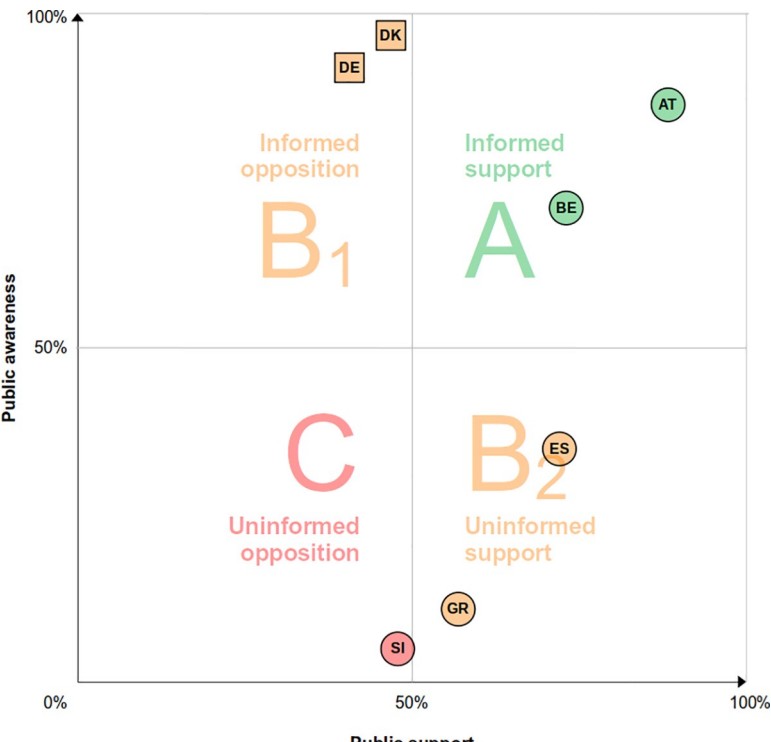

**Fig 3. National prevalence of policy awareness against support.** Squares represent opt-in countries, and circles represent opt-out countries.

consent policy (Table 7 and Fig B in S1 File). A χ2 test revealed that rates of preference expression varied markedly across countries, $\chi^2(df = 6) = 290.8$, $p < .001$ (see also Table 6).

To examine differences in expression rates across consent systems, we regressed expression on consent system, while treating country as a random effect. This mixed-effects logistic regression revealed a significant effect of consent system on expression: Specifically, expression rates were dramatically higher in opt-in (83%, 95% CI [71%, 90%]) than in opt-out (47%, 95% CI [37%, 58%]) countries, OR = 5.32, 95% CI [2.39, 11.85], $z = 4.09$, $p < .001$.

**Table 6. Risk of contravening non-expressed preferences: Total frequency, and convergence with default, by country.**

| System | Country | Non-expression n (% total) | Non-expressed personal preference and policy default n (% total) | | |
|--------|---------|---------|---------|---------|---------|
| | | | Convergent | Don't know | Divergent |
| Opt-in | DE | 56 (13) | 13 (3) | 22 (5) | 21 (5) |
| | DK | 53 (23) | 3 (1) | 24 (10) | 26 (11) |
| Opt-out | AT | 171 (50) | 149 (44) | 15 (4) | 7 (2) |
| | BE | 201 (46) | 162 (37) | 30 (7) | 9 (2) |
| | ES | 78 (35) | 64 (29) | 12 (5) | 2 (1) |
| | GR | 96 (60) | 58 (36) | 27 (17) | 11 (7) |
| | SI | 138 (73) | 85 (45) | 37 (20) | 16 (8) |

*Note.* Table 6 depicts the risk national consent policies face of contravening ("divergent" column) vs encompassing ("convergent" column) individuals' unexpressed preferences. The "don't know" column refers to individuals who simultaneously have not (officially) expressed a preference and who declared to be undecided about their preference regarding *post-mortem* OP. In parenthesis, percentages are calculated with the total national sample size. These figures do not take into account whether or not individual non-expression is autonomous (e.g., whether individuals are aware of the policy in place and the implications of remaining silent).

**Table 7. Levels of governance quality: Respect for policy-dissenters unexpressed preferences (by country) and policy awareness.**

| System | Country | Levels | | | |
|---|---|---|---|---|---|
| | | A' | B$_1$' | B$_2$' | C' |
| | | *Informed convergence* | *Informed divergence* | *Uninformed convergence* | *Uninformed divergence* |
| Opt-in | DE | 12 (3) | 18 (4) | 1 (0) | 3 (1) |
| | DK | 3 (1) | 26 (11) | 0 (0) | 0 (0) |
| Opt-out | AT | 133 (39) | 5 (1) | 16 (5) | 2 (1) |
| | BE | 105 (24) | 4 (1) | 57 (13) | 5 (1) |
| | ES | 20 (9) | 0 (0) | 44 (20) | 2 (1) |
| | GR | 6 (4) | 1 (1) | 52 (33) | 10 (6) |
| | SI | 2 (1) | 1 (1) | 83 (44) | 15 (8) |

*Note*. Table 7 assesses the risk of individuals' preference contravention due to policy unawareness. By taking into account whether or not individual non-expression is informed or uninformed (whether or not individuals know the consent policy in place), these figures measure national policies' capacity to respect/contravene the autonomy of policy-dissenting individuals. In parenthesis, percentages are calculated with the total national sample size.

Additionally, by juxtaposing policy awareness and fulfilment of unexpressed preferences, we can compare where the analyzed countries stand in relation to each other. Fig 4 shows, for each country, the proportion of participants who know the policy in place (vertical axis) and the proportion of individuals whose unexpressed preferences are likely to be fulfilled because they converge with the policy standard in the country (horizontal axis).

Austria and Belgium are again situated in the best-case scenario A' (informed convergence): most silent participants know their national policy, suggesting that, when remaining silent,

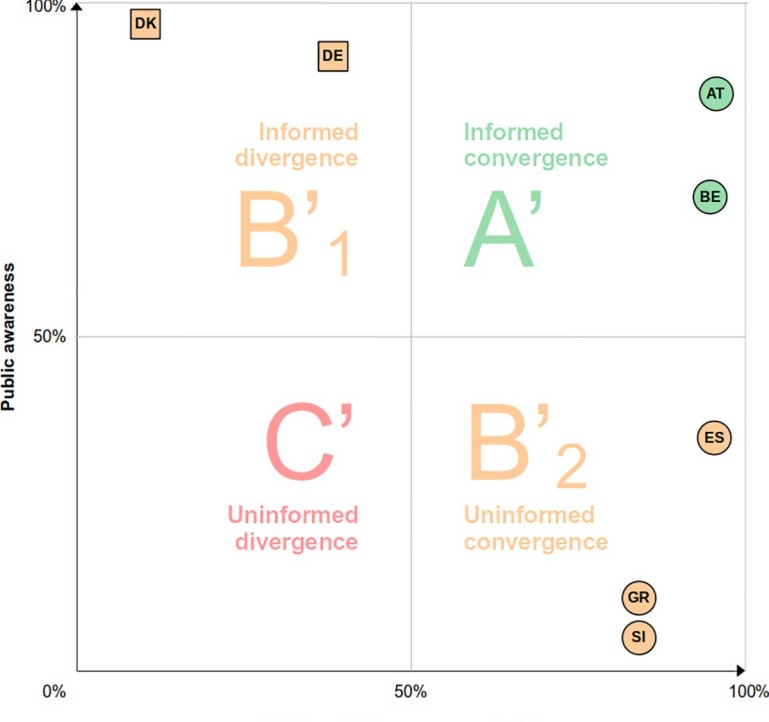

**Fig 4. National prevalence of policy awareness against fulfilment of non-expressed preferences.**

they make autonomous decisions which will eventually lead to the fulfilment of their preference (to donate). In Denmark and Germany, most individuals know the opt-in policy, suggesting that their silence reflects an autonomous choice as well. However, their unexpressed preferences (to donate) diverge from the policy default and will likely not be fulfilled, thus situating these two countries in the suboptimal scenario $B'_1$ (informed divergence). In Greece, Slovenia and Spain, only a minority of respondents know the opt-out policy, suggesting that their decision is not the result of an autonomous choice. Notwithstanding, their non-expressed preference will likely be fulfilled–as their wish to donate converges with their national opt-out policies–thus situating these countries in scenario $B'_2$ (uninformed convergence). Notably, no country is situated in scenario C (uninformed divergence).

## Discussion

Organ donation policy-making deals with competing interests and values regarding the common good, individual preferences, and public trust in institutions. Each system allows individuals to express their donation preferences–either to consent or to refuse organ procurement, or both [21]–by registering their decision, by holding an organ donor/non-donor card, by writing down advanced directives, or by communicating their preferences to their family members. In most cases, the expressed wishes of the deceased (in favor or against donation) are respected, regardless of the consent system. Indeed, individuals' explicit refusal to donate, however it is expressed, will likely be respected in both opt-in and opt-out countries (following national and international laws and ethical guidelines [8]. Likewise, an explicit consent to donate will most likely be respected under both opt-in and opt-out policies (even though relatives may be allowed to overrule or *veto* the deceased's decision under any of those two policies) [5]. Therefore, the real difference between opt-in and opt-out policies does not rely on the deceased's decision–i.e., consent or refusal–but on the default course of action that applies *when the deceased failed to express any decision*: organs can be recovered under opt-out policies and they cannot be recovered under opt-in policies.

By setting a default course of action, opt-out and opt-in policies facilitate dealing with the aforementioned tensions, by assisting clinicians to decide whether or not the organs of those who have not expressed any preference regarding organ donation should be procured. However, by doing so they may also misrepresent or violate the interests of some. Ideally, such policies should encompass the values and preferences of an informed majority, and minimally compromise the values and preferences of those who dissent from the default course of action settled by the policy. In practice, a number of circumstances may hinder such ideal governance requirements, including people holding different views regarding OP, people failing to express their preferences, people being unaware of the enforced policy, and people disagreeing with it. Under such circumstances, consent policies for OP unavoidably incur in the risk of misrepresenting the values of some parts of the society and contravening the legitimate preferences of some individuals. This paper tests a model for measuring that moral and political risk based on peoples' preferences regarding OP, their knowledge of the consent policy in place in their countries, and their degree of support of such policies.

Our results show that a majority of participants in all countries express a preference in favor of organ donation and an opt-out model for *post-mortem* organ retrieval. These results suggest general consistency between public interests and the international regulatory trend to move towards opt-out models of consent. Still, different quality levels of OP policy governance appear across countries, fundamentally due to different levels of public awareness of consent policies, different levels of public endorsement of such policies, and different levels of individual expression of personal preferences regarding OP. Do these results suggest different

governance quality assessments in opt-in and opt-out countries? We first discuss our results from the perspective of each policy's capacity to represent the values of an informed majority, then we consider their capacity to preserve the values of dissenting individuals.

## Consent policies' capacity to represent the preference of the majority

Participants in the two opt-in countries included in this study–Denmark and Germany– revealed overall lower policy endorsement but higher levels of policy awareness than participants in any other country. As most Danish and Germans simultaneously know their system but would prefer opt-out, their governance quality score corresponds with the suboptimal level $B_1$ that we have characterized as *informed opposition*. Further empirical evidence is needed to confirm this result, in which case OP policy-making in these countries should consider moving to opt-out to better accommodate the preferences of the majority.

In contrast, the public in opt-out countries tend to support their consent system more and to know it less. To a great extent, this result is consistent with previous evidence on public awareness of consent policies for OP [17] which may reflect increased efforts in opt-in countries to inform the population through public campaigns to motivate donation [22]. In our sample, low levels of public policy awareness particularly appear in Greece and Spain, whose governance quality scores correspond to suboptimal level $B_2$, *uninformed support*. Organ procurement policy-making in these countries should consider increasing public awareness of the opt-out policy in place.

By combining high support with considerably high awareness of the presumed consent system, Austria and Belgium score higher than any other country (opt-out or opt-in) in this measure of governance quality (level A: *informed support*). These two opt-out countries meet the conditions for policy stability, as a majority of respondents are both familiar and supportive of their regulation. In the opposite side, by combining low support and low awareness of the presumed consent system, Slovenia scores the lowest of the seven countries reviewed and is situated at level *C*, *uninformed opposition*. If confirmed by nationally representative data, policy-making in this country may consider either increasing public awareness of opt-out or shifting to opt-in (although this option seems less advisable, as 48% of participants in Slovenia support opt-out).

## Consent policies' capacity to preserve the values of policy-dissenting individuals

Our results show that, in opt-in countries, individual preferences regarding OP are more often expressed, as compared to opt-out countries. This factor accounts for opt-in countries only exceptionally facing the risk of contravening the preferences of policy-dissenting individuals. Besides, in Denmark and Germany, only a minority is unaware of the opt-in policy in place. This suggests that, in the few instances where the interests of the silent minority (who fail to register their preference to donate) can be violated, not procuring the organs may still be consistent with peoples' autonomy (as they have presumably chosen not to register their preference). The governance quality score in these countries corresponds to the suboptimal level $B'_1$ that we have characterized as *informed divergence*. Such outcome can be considered as unproblematic from the moral perspective of respecting individual values, as decisions are autonomous after all.

In opt-out countries, policies' capacity to preserve the values of policy-dissenting individuals is more diverse. Spain, Greece and Slovenia, on the one hand, combine medium to low level of public policy awareness with high to very high levels of fulfilment of non-expressed preferences (high levels of preference-policy convergence). This results in suboptimal $B'_2$

governance quality scores (*uninformed convergence*). In these countries, policy violation of the values of policy-dissenting individuals (who may have their organs procured despite refusing organ donation) are also unlikely. However, given this populations' unawareness of the opt-out policy in place, such violations, as exceptional as they might be, should be considered disrespectful with their autonomy. Importantly, in most countries, the family of the deceased are allowed to make decisions over OP when the deceased person had not [5]. This may reduce the risk of actual violations of peoples' autonomy to anecdotal incidence. Still, policy makers may consider alternative ways, including increased publicity about opt-out, to ensure that people who refuse OP, most of whom ignore their automatic status as donors, will eventually have their preference respected.

Austria and Belgium, on the other hand, combine high levels of preference-policy convergence with high levels of policy awareness. Their corresponding governance quality score again is the optimal level A' (informed convergence), meaning that these systems optimally preserve the preferences of the individuals who would not want to donate and that, if such an event ever happened, it would likely be the result of an autonomous choice.

A promising result of our study is that none of the seven participant countries matched the C' category of uninformed divergence, and that only one (Slovenia) scored the lowest governance quality C regarding uninformed opposition.

## Limitations

This study has some limitations. First, the surveyed population–university students–is not representative of the general population on key variables such as age, educational attainment, gender and socioeconomic status. Reassuringly, however, previous studies on knowledge and attitudes toward organ donation policies in Europe show no significant effects of age or gender on awareness of organ donation laws [23], willingness to donate their own organs after death or organs from a deceased close family member [23, 24], and no age differences in willingness to donate tissues after death [25]. Still, education level has been found to predict donation-related attitudes: enrolment in tertiary studies is associated with greater awareness of the law and support for posthumous organ or tissue donation [23–25]. Therefore, we can expect that nationally representative studies would show lower levels of policy awareness, resulting in lower national scores in policy governance quality assessments. However, the education bias can be seen as less problematic for our approach, as we were interested in the relation of attitudes and knowledge, and less on absolute knowledge levels.

Second, our sample shows a gender bias across all countries and study fields. The fact that more women took part in the survey than did men can be partially explained by high female-to-male ratios in the sampling frame (S1 File)–particularly among majors in health sciences (54–72%; vs. 39–62% for humanities/social sciences)–as well as in the European student population (72% for health sciences students; 64% for humanities/arts/social sciences students; see [18]. Controlling for the gender distribution in the population, we still observe some overrepresentation of women in our sample, which may stem from sex differences in non-response bias, previously detected in other studies [26] indicating higher retention and completion rates among women than men. Third, in comparison to other surveys about organ donation [27], the present questionnaire was longer and more time-consuming, so this can also select for highly motivated participants. This reason may account for some students' decision not to participate in the study. Fourth, this study assesses the quality of OP policies by focusing only on their capacity to preserve and promote societal values. It does not address the question of the effectiveness of presumptive policies, which has been extensively explored elsewhere [28, 29]. As health policies attempt to foster optimal results while minimally compromising people's

interest, future research that combines both approaches would be much welcomed. Fifth, our model for governance quality assessment omits the involvement of potential donors' families in the decision-making process. Families may in fact determine whether or not default policies are followed, and the extent to which the autonomy of the deceased is respected. However, the level of authority given to families in each jurisdiction (*de iure*) can be ambiguous, and is often inconsistent with the role they actually play in practice (*de facto*) [5, 30], thus precluding the use of this dimension in our model of governance, which is based on societal awareness of actual policies. Finally, our study does not address the theoretical controversy about the relevance of posthumous interests, which has been both disputed [31] and supported [32] on philosophical grounds.

Despite these limitations, by showing the moral and political relevance of connecting peoples' knowledge, preferences, and their respective policy setting, our model provides more information for an ethics assessment of current OP policies than most public surveys [17] conducted on public attitudes in this health domain. Our research proposes and tests a new theoretical approach to assess health governance of international OP policies. This theoretical framework can serve as a basis to measure health governance quality regarding OP in any single country, or to compare it across countries. Future work should use nationally representative population data to conduct the same governance analysis. The defined governance scores can be used retrospectively on results from previous studies, provided that they include the necessary variables: knowledge of the policy, support for the policy, willingness to donate, and rates of preference expression.

## Conclusions

Organ procurement is morally contentious, in part because it pits people's exercise of bodily autonomy against the public health interest in recovering organs, and ultimately, in promoting aggregate health. Deceased organ procurement policy-making seeks to minimize this conflict in two ways: (1) by maximizing support for the policy in place, and (2) minimizing violations of dissenters' posthumous preferences. This study develops and tests a framework to assess the health governance of consent policies for OP, based on people's knowledge and support of the enforced policy in their countries, their preference regarding OP, and their preference expression. We think this framework can be used for other settings, including other areas in health care where individual preferences are presumed or cannot be always reconciled with public interest. The current debates about COVID19 pandemic measures such as physical distancing and self-restriction in mobility, or vaccination policy for serious diseases would be other examples.

Across seven European countries, we observe high rates of willingness to donate and a general preference for presumed consent policies. Our results also show an increased tendency for people in opt-in countries to be aware of the enforced policy, which, coupled with their willingness to donate, could also explain the high levels of preference expression—relative to opt-out countries. As a result, the risk of contravening the preferences of policy dissenters in opt-in countries—i.e., of individuals who *would* like to donate—is minimal.

Meanwhile, examined opt-out countries enjoy broad policy support (with Slovenia being an exception) but exhibit far greater variability in both policy awareness and preference expression. When policy awareness is high, as in Austria and Belgium, governance quality is the highest—as most citizens in these countries also favor the policy in place. In this context, preference non-expression poses a negligible ethical risk. In contrast, when policy awareness is low, in Spain, Greece and especially Slovenia, governance quality is compromised. People's failure to express a preference could stem from a misunderstanding of the consequences of

non-expression. Luckily though, most individuals in these countries also report a preference for *post-mortem* OP; therefore, posthumous preferences in these countries are also likely to be satisfied—even when they go unexpressed.

Different strategies may serve to improve governance quality in each country, depending on their predominant profile. Opt-in countries, where people are largely informed but unsupportive, could seek greater governance quality by shifting to people's preferred consent system, i.e., opt-out (under the assumption that these preferences are relatively rigid). In turn, opt-out countries, where people are generally supportive but insufficiently informed, could pursue greater governance quality by educating citizens about the existing consent system and the means through which to communicate any preferences regarding OP. Of course, maximizing awareness would also come at some detriment—namely, to overall organ availability—by increasing the likelihood that dissenters will exercise their right to abstain from donation.

## Supporting information

**S1 File.**
(DOCX)

## Acknowledgments

We would like to thank all members of the ISSATO working group for data collection coordination: Gabriele Werner-Felmayer and Magdalena Flatscher-Thöni (Austria), Kristof van Assche (Belgium), Anja Jensen (Denmark), Thalia Bellali (Greece), and Tanja Kamin (Slovenia). We are also thankful to other members of the ELPAT-ESOT (Ethical, Legal and Psychosocial Aspects of Transplantation-European Society of Organ Transplantation) for their help in shaping the questionnaire: Hagai Boas, Myfanwy Morgan, and Gurch Randhawa. Yolanda Ramallo Fariña and Miguel Angel García Bello helped us with methodological advice. Finally, we thank ELPAT-ESOT board for organizing working-group meetings between 2016 and 2018.

## Author Contributions

**Conceptualization:** David Rodríguez-Arias, Alberto Molina-Pérez, Janet Delgado, Sabine Wöhlke, Silke Schicktanz.

**Data curation:** Benjamin Söchtig.

**Formal analysis:** Alberto Molina-Pérez, Ivar R. Hannikainen.

**Funding acquisition:** David Rodríguez-Arias.

**Investigation:** David Rodríguez-Arias, Alberto Molina-Pérez, Janet Delgado, Benjamin Söchtig, Sabine Wöhlke, Silke Schicktanz.

**Methodology:** David Rodríguez-Arias, Alberto Molina-Pérez, Ivar R. Hannikainen, Janet Delgado, Sabine Wöhlke, Silke Schicktanz.

**Project administration:** Sabine Wöhlke, Silke Schicktanz.

**Resources:** Benjamin Söchtig.

**Supervision:** Sabine Wöhlke.

**Visualization:** Alberto Molina-Pérez, Ivar R. Hannikainen.

**Writing – original draft:** David Rodríguez-Arias, Alberto Molina-Pérez, Ivar R. Hannikainen, Janet Delgado.

**Writing – review & editing:** David Rodríguez-Arias, Alberto Molina-Pérez, Ivar R. Hannikainen, Janet Delgado, Benjamin Söchtig, Sabine Wöhlke, Silke Schicktanz.

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
