## [Decision Letter · Decision Letter 0]

15 Feb 2021

PONE-D-21-01301

Governance Quality Indicators for National Organ Procurement Policies. A novel approach based on a Cross-European Survey of Students’ Knowledge and Views about Consent Policies

PLOS ONE

Dear Dr. Rodriguez-Arias,

Thank you for submitting your manuscript to PLOS ONE. After careful consideration, we feel that it has merit but does not fully meet PLOS ONE’s publication criteria as it currently stands. Therefore, we invite you to submit a revised version of the manuscript that addresses the points raised during the review process.

Interesting and relevant paper with an original and thought-provoking analysis. Expert reviewers have come up with some concerns (details below) on the the validity of the governance analysis and the sample being clearly highly selective. These shortcomings need to be acknowledged. Also, the presentation of the statistical results and justification of the sample are not convincing. Other comments and concerns are detailed below. Please revise the MS accordingly, including a point-by-point response to the reviewers' comments. Please be aware that the invitation to resubmit your work with major revisions does not imply acceptance following that, as the paper will undergo rigorous peer review again. Looking forward to the revised MS.

We look forward to receiving your revised manuscript.

Kind regards,

Frank JMF Dor, M.D., Ph.D., FEBS, FRCS

Academic Editor

PLOS ONE

Journal Requirements:

2. Thank you for including your ethics statement:  "University of Göttingen Human Research Review Committees Ref. Nr. 13/01/19, and Human Research Review Committee and the Universidad de Granada Ethics Committee on Human Research".   

Please amend your current ethics statement to confirm that your named institutional review board or ethics committee specifically approved this study.

4. During our internal checks, the in-house editorial staff noted that you conducted research or obtained samples in another country. Please check the relevant national regulations and laws applying to foreign researchers and state whether you obtained the required permits and approvals. Please address this in your ethics statement in both the manuscript and submission information. In addition, please ensure that you have suitably acknowledged the contributions of any local collaborators involved in this work in your authorship list and/or Acknowledgements. Authorship criteria is based on the International Committee of Medical Journal Editors (ICMJE) Uniform Requirements for Manuscripts Submitted to Biomedical Journals - for further information please see here: https://journals.plos.org/plosone/s/authorship.

5. Please note that in order to use the direct billing option the corresponding author must be affiliated with the chosen institute. Please either amend your manuscript to change the affiliation or corresponding author, or email us at plosone@plos.org with a request to remove this option.

6. Please amend your list of authors on the manuscript to ensure that each author is linked to an affiliation. Authors’ affiliations should reflect the institution where the work was done (if authors moved subsequently, you can also list the new affiliation stating “current affiliation:….” as necessary).

Reviewers' comments:

Reviewer's Responses to Questions

**Comments to the Author**

1. Is the manuscript technically sound, and do the data support the conclusions?

Reviewer #1: Partly

Reviewer #2: Yes

Reviewer #3: Partly

2. Has the statistical analysis been performed appropriately and rigorously? 

Reviewer #1: I Don't Know

Reviewer #2: Yes

Reviewer #3: No

3. Have the authors made all data underlying the findings in their manuscript fully available?

Reviewer #1: Yes

Reviewer #2: Yes

Reviewer #3: Yes

4. Is the manuscript presented in an intelligible fashion and written in standard English?

Reviewer #1: Yes

Reviewer #2: Yes

Reviewer #3: Yes

5. Review Comments to the Author

Reviewer #1: This is an interesting study addressing a fundamental problem in organ procurement for transplantation, namely attitudes and education. The study has a number of strengths. It is well written, in good English, and well reasoned. The number of respondents is large although there is clearly a selection bias in terms of the demographics of respondents. The questionnaire was comprehensive and well researched. The conclusions are interesting and thought provoking, describing a novel form of governance. I would raise the following observations:

Major

The authors propose a novel tool for governance and then claim that their study, "validates a tool for analysing policy governance....". I do not believe that the tool has been validated as there is no standard to validate against. This need further discussion. and a proposal as to how it might be validated.

The respondent population is highly selective which is acknowledged towards the end of the manuscript but this needs more prominence. While the findings are interesting to be conclusive this finding needs to be replicated in a more unbiased population than largely female students in their twenties in large metropolitan universities. This needs more discussion.

Despite the discussion of default options there should be some discussion addressing the difference between "soft" and "hard" default options whereby next of kin can override donor's overtly expressed wishes.

I would defer to a statistician regarding the statistical analysis

Minor

Introduction - the last word in the first paragraph should be "latter".

Page 22 - there is no such verb as "concept-proves"

Reviewer #2: This is an interesting study which is very well performed and gives insight in knowledge and views about consent policies in seven countries. The authors created a novel tool for analyzing governance quality and illustrates how the strengths and weaknesses of different policy implementations can be estimated and compared using quantitative survey data.

I only have a few details I would like to comment on:

1. In this study countries are divided into 'opt-out' or 'opt-in' consent system. However, this is not always black and white, for example Belgium (opt-out) system has the possibility to register 'objection' AND 'consent' to donation. The number of consent registration even doubles the registration of objection. This nuance is relevant for the 'limitations of the study' section.

2. A minor detail is the use of the words 'organ procurement', what is the reason that the authors do not use 'organ donation'? Procurement reflects the procedure in the operating room.

3. A second minor detail is in the Discussion section, which starts with 'Organ transplantation policy-making...', I think the authors mean 'Organ donation policy-making'.

4. On page 24 it says “Importantly, in most countries, the family of the deceased are allowed to make decisions over OP when the deceased person had not [5].” In practice donation is always discussed with the family, regardless an opt-out or opt-in system. Several studies have shown that family sometimes overrules donor preferences.

Reviewer #3: The statistical analysis approach and software appear reasonable for this research. However there are concerns with the sample and the general presentation of the results.

1. The investigators claim to have a sample of 2006. Why is there a high percentage of females? What is the general size of the population from which the sample is drawn and how representative is this sample of the population and in particular of the opt in and opt out countries as well as the disciplines being sampled? What statistical plan was in place to determine that 2006 was, in fact, the sample needed for this survey?

2. What bias checks were made of the sample to assume its validity in this context?

3. The investigators use p-values in general. However, they use terms or expressions such as ‘preponderance’ and ‘tendency to support’. What is the quantitative setting for these terms?

6. PLOS authors have the option to publish the peer review history of their article (what does this mean?). If published, this will include your full peer review and any attached files.

Reviewer #1: No

Reviewer #2: No

Reviewer #3: No

---

## [Author Response · Author response to Decision Letter 0]

22 Apr 2021

Response to reviewers. 

Please find attached a new version of the manuscript “Governance Quality Indicators for National Organ Procurement Policies”. This new version addresses the questions raised by the reviewers, as detailed below. Additionally, we have introduced unrequested minor modifications to improve figures display and wording. We appreciate the comments received, which gave us the opportunity to improve the original manuscript. 

Reviewer #1: 

“This is an interesting study addressing a fundamental problem in organ procurement for transplantation, namely attitudes and education. The study has a number of strengths. It is well written, in good English, and well reasoned. The number of respondents is large although there is clearly a selection bias in terms of the demographics of respondents. The questionnaire was comprehensive and well researched. The conclusions are interesting and thought provoking, describing a novel form of governance. I would raise the following observations”:

We appreciate Reviewer 1’s kind words about our manuscript. In the revised submission, we have taken numerous steps to quantify the extent of gender imbalance, examine its sources, and have also emphasized the limitations inherent to our convenience sampling approach. 

Major

The authors propose a novel tool for governance and then claim that their study, "validates a tool for analyzing policy governance....". I do not believe that the tool has been validated as there is no standard to validate against. This needs further discussion and a proposal as to how it might be validated.

As Reviewer 1 notes, we did not assess the proposed tool’s validity or reliability. We acknowledge that referring to our study as ‘validating’ the tool was misleading, as there is no previous measurement of governance quality with which our results can be compared. In the revised manuscript, we have replaced the term “validates” with the unambiguous alternative “introduces”, e.g., in the Abstract:

(Abstract) 

→ “Our study ‘introduces’ a tool for analyzing policy governance”).

The respondent population is highly selective which is acknowledged towards the end of the manuscript but this needs more prominence. While the findings are interesting to be conclusive this finding needs to be replicated in a more unbiased population than largely female students in their twenties in large metropolitan universities. This needs more discussion.

In the revised submission, we have clarified that we used a convenience sample and have given more attention to the overrepresentation of women in our sample. 

We clarify the recruitment method in the Abstract and in the Introduction, as follows: 

(Abstract) 

→ Methods: Between 2017-2019, we conducted a convenience sample survey of students (n=2006) 

(Introduction; Page 5)

 → “We do so by administering a survey to a convenience sample of university students focusing on […]”

In preparation for our resubmission, we also conducted new analyses attempting to diagnose the causes of such stark gender imbalance in our sample. As a result of these analyses, we now claim in the paper that the high female-to-male ratio in the study sample is due, to a certain extent, to the greater proportion of women already present (a) in the sampling frame (i.e., the study programs from which we recruited our sample), and also (b) in the population (i.e., tertiary education in Europe as a whole). 

To support this, we first provide detailed information of the student gender distribution in the sampling frame: i.e., for each participating university (when available) and study field (health sciences on the one hand, and humanities, social sciences and arts on the other) on the year when data collection took place. This information is made available in the Supporting Information file as well as in the bottom of this cover letter. Second, we provide more general information about gender distribution in the population: i.e., drawing from Eurostat census information on tertiary education throughout Europe. These explorations suggest that the gender imbalance in our sample is not as pronounced as it might initially appear: The female-to-male ratio is 2.85:1 in our sample, and 2.15:1 in the population. In the revised manuscript, we suggest that the remaining overrepresentation (relative to the population) can be accounted for by known sex differences in non-response bias, since response rates have been found to be higher among women than among men –at least for volunteer-based, survey studies (REFERENCE ADDED). We highlight this potential limitation in the Introduction (p6), in the Methods section (p12), and in the Discussion (p25), as follows: 

(Introduction Pages 5-6)

 → “Our sample (n=2006) serves the purpose of testing the theoretical framework, but the respondent population is highly selective and cannot be considered representative of national populations, as discussed below”. 

(Methods, Page 12)

→ “Overall, 2006 Austrian, Belgian, Danish, German, Greek, Slovenian and Spanish students (age bracket (mode) = 20 to 24 years) took part in our study. A majority of them (74%) were women (see Table 1), which partially reflects an overrepresentation of women in the disciplines concerned [Supplementary file Table 1 ADDED] [REFERENCE EUROSTAT ADDED], but may also account for an answer bias, as mentioned in the Discussion section”.

(Discussion, Page 25)

 → “Second, our sample shows a gender bias across all countries and study fields. The fact that more women took part in the survey than did men can be partially explained by high female-to-male ratios in the sampling frame (supplementary file) –particularly among majors in health sciences (54-72%; vs 39-62% for humanities/social sciences)– as well as in the European student population (72% for health sciences students; 64% for humanities/arts/social sciences students; see [EUROSTAT, 2018]). Controlling for the gender distribution in the population, we still observe some overrepresentation of women in our sample, which may stem from sex differences in non-response bias, previously detected in other studies [19, REFERENCE ADDED Damman et al] indicating a higher retention and completion rates among women than men.” 

We agree with Reviewer 1 that future work should conduct the same governance analysis on the basis of representative panel data, as we mention in the Discussion (p27):

(Discussion, Page 26)

→ “Future work should use nationally representative population data to conduct the same governance analysis”.

Despite the discussion of default options there should be some discussion addressing the difference between "soft" and "hard" default options whereby next of kin can override donors’ overtly expressed wishes.

Reviewer 1 is right that models of consent for organ procurement do not rely on default options alone, but critically depend on the authority given to families as well. Consent systems for organ recovery are indeed more complex than just default options, as their implementation also depends on the available procedures to express individual preferences and, even more importantly, on the authority given to the next of kin. This is partially acknowledged in the introduction (p.2) and in the discussion section (p.25), where we claim that: 

“Consent policies for post-mortem OP establish how citizens may express their preferences (e.g., donor and refusal registries, donor cards, etc.), next-of-kin’s role in OP decision-making, and the default rule (retrieving or not retrieving the organs) when no preference is available. Jurisdictions vary regarding the means of expressing individuals’ preferences and the authority of the family [5]” (p2)

“in most countries, the family of the deceased are allowed to make decisions over OP when the deceased person had not [5]. This may reduce the risk of actual violations of peoples’ autonomy to anecdotal incidence”. (p25)

Reviewer 1’s comment also underlies the distinction between “soft” consent systems, (where family preferences are taken into account and are allowed to interfere with the default option) and “hard” consent systems (where the default policy is applied regardless of family preferences). We acknowledge that this distinction could be relevant for our policy governance assessment. On the one hand, family involvement in the decision-making process may determine the extent to which individual preferences and the autonomy of the deceased is respected. On the other hand, individuals may fail to express their preferences regarding organ retrieval if they believe their families will eventually have a say in the decision, or even the last word. 

Our model for policy governance assessment uses awareness of, and agreement with the policy in force as indicators for governance quality. Unfortunately, this type of policy analysis is not applicable, in the current state of knowledge, for family involvement in decision making, for the following reasons:

To include the family dimension in our governance quality model, we would have had to explore participants’ awareness of the role family plays in their country, and their level of agreement with such policy. However, reliable information about the role families play in each country is either ambivalent, contradictory, or simply not available, thus making it impossible for us to assess whether individuals know it. In a previous work, we have argued that the exact role families play in each jurisdiction can hardly be captured by the twofold “hard/soft” dichotomy (Delgado et al 2019): In fact, families may have four incremental levels of involvement in the organ retrieval decision-making process: 1. they may have no role whatsoever, 2. they may only be expected to witness or update the deceased’s preferences, 3. they may act as surrogate decision-makers and decide on behalf of the deceased, and finally, 4. they may have full decisional authority, meaning that they could override both a consent, and a refusal (although there is no robust evidence that the latter ever happens). Even more crucially for the purpose of including family decision-making in our governance model, the classification of countries according to these four levels (or, for the same purposes, according to the hard/soft dichotomy) depends on whether the focus is placed on the authority families are given by law (de iure) or the authority they actually have in practice (de facto). However, two of the authors have shown that, in most countries, there is no correspondence between the official and the actual role families play, the latter being only supported by partial and unreliable data (Morla et al, forthcoming). To give a few examples, in Spain, Chile, and to some extent in Austria, the law only gives families the authority to witness or update the deceased’s preferences, but families may actually be allowed by doctors to have the last word (meaning that they can in practice overrule the deceased’s consent to donate) (Delgado et al. 2019). Another example is France, where a recent law has been enacted to avoid family veto, but there is evidence suggesting that doctors still prefer not to procure the organs of some potential donors when they perceive that doing so could undermine the relationship of trust they have with their families (Touraine 2017). These inconsistencies between law and practice create policy indetermination which precludes a proper policy governance analysis based on policy knowledge: participants may be simultaneously right and wrong if they believe that families hold a responsibility in organ procurement decision-making. 

As a matter of fact, our survey attempted to address this topic by including two questions that explored participants' beliefs about the role given to families when the deceased had expressed a preference, and when they had not (see Supplementary File, survey Questions 6 and 7). However, the team decided that ambiguity about policy implementation in this respect recommended to leave these results out from our policy governance model, as they were difficult to interpret and any conclusion on family decision-making would be misleading. More tailored research on family involvement as part of policy governance quality assessments seems warranted, which should take into account the law/practice inconsistencies we have identified. Although we cannot address in depth this issue in this manuscript, we agree with Reviewer 1 (and with Reviewer 2) that the role played by the next of kin deserves further recognition in our paper. 

We have introduced a clarification on the role families play at the beginning of the discussion section: 

(Page 20)

→ Indeed, individual’s explicit refusal to donate, however it is expressed, will likely be respected in both opt-in and opt-out countries (following national and international laws and ethical guidelines) [REFERENCE ADDED WHO 2010]. Likewise, an explicit consent to donate will most likely be respected under both opt-in and opt-out policies, (even though relatives may be allowed to overrule or veto the deceased's decision under any of those two policies) (REFERENCE ADDED Delgado et al. 2019).”

Additionally, we have acknowledged this limitation of our model. (Discussion section),

(Pages 25-26)

→ “Fifth, our model for governance quality assessment omits the involvement of potential donors’ families in the decision-making process. Families may in fact determine whether or not default policies are followed, and the extent to which the autonomy of the deceased is respected. However, the level of authority given to families in each jurisdiction (de iure) can be ambiguous, and is often inconsistent with the role they actually play in practice (de facto) (REFERENCES ADDED Delgado et al 2019; Morla et al.), thus precluding the use of this dimension in our model of governance quality, which is based on societal awareness of actual policies”. 

Minor

Introduction - the last word in the first paragraph should be "latter".

This has been corrected: 

(Page 2)

→ “a novel framework of analysis that focuses on the latter”

Page 22 - there is no such verb as "concept-proves"

This has been corrected:

(Page 21)

→ “This paper tests a model for measuring that moral and political risk based on peoples ...”

 

Reviewer #2: 

This is an interesting study which is very well performed and gives insight in knowledge and views about consent policies in seven countries. The authors created a novel tool for analyzing governance quality and illustrates how the strengths and weaknesses of different policy implementations can be estimated and compared using quantitative survey data.

I only have a few details I would like to comment on:

1. In this study countries are divided into 'opt-out' or 'opt-in' consent system. However, this is not always black and white, for example Belgium (opt-out) system has the possibility to register 'objection' AND 'consent' to donation. The number of consent registration even doubles the registration of objection. This nuance is relevant for the 'limitations of the study' section.

Reviewer 2’s suggestion has made us realize the need to further clarify the difference between opt-in and opt-out. We do so at the beginning of the Discussion section (p.21), where we mention that some systems may also allow individuals to express both a consent and a refusal. We also insert a reference (Rosenblum et al 2012), where this is explicitly acknowledged: 

(Page 20)

→ “Each system allows individuals to express their donation preferences –either to consent or to refuse organ procurement, or both (see Rosenblum et al. 2012)– by registering their decision, by holding an organ donor/non-donor card, by writing down advanced directives, or by communicating their preferences to their family members. In most cases, the expressed wishes of the deceased (in favor or against donation) are respected, regardless of the consent system. Indeed, individuals’ explicit refusal to donate, however it is expressed, will likely be respected in both opt-in and opt-out countries (following national and international laws and ethical guidelines, such as WHO 2010). Likewise, an explicit consent to donate will most likely be respected under both opt-in and opt-out policies, (although relatives may be allowed to overrule or veto the deceased's decision under any of those two policies) (Delgado et al. 2019). Therefore, the real difference between opt-in and opt-out policies does not rely on the deceased’s decision –i.e. consent or refusal– but on the default course of action that applies when the deceased failed to express any decision: organs can be recovered under opt-out policies and they cannot be recovered under opt-in policies.”

3. A second minor detail is in the Discussion section, which starts with 'Organ transplantation policy-making...', I think the authors mean 'Organ donation policy-making'.

 We agree, and have changed the wording accordingly 

(Page 20) 

→ “Organ donation policy-making deals with competing interests and values regarding the common good, individual preferences, and public trust in institutions”

4. On page 24 it says “Importantly, in most countries, the family of the deceased are allowed to make decisions over OP when the deceased person had not [5].” In practice donation is always discussed with the family, regardless an opt-out or opt-in system. Several studies have shown that family sometimes overrules donor preferences.

This comment by Reviewer 2 converges with the comment made by Reviewer 1 on the relevance of the difference between “soft” and “hard” consent systems. A full response to that comment is provided above. Reviewer 2 is right: in practice, in most countries, doctors give families the authority to overrule donors’ preferences (especially when they express the wish to donate), even though the law in their jurisdiction may not grant them such responsibility. As mentioned above, including the role of families in our model of governance quality assessment was beyond our capacity, because the exact role families play is surrounded by epistemic uncertainty. Notwithstanding, the corrected version of the manuscript stresses the fact that families may trump the default option, thus turning a particular consent system ineffective. We have introduced some changes in the discussion section: 

(Page 20)

 → “an explicit consent to donate will most likely be respected under both opt-in and opt-out policies, (even though relatives may be allowed to overrule or veto the deceased's decision under any of those two policies) (REFERENCE ADDED Delgado et al. 2019)”.

(Page 25)

→ “Fifth, our model for governance quality assessment omits the involvement of potential donors’ families in the decision-making process, even though families may in fact determine whether or not default policies are followed, and the extent to which the autonomy of the deceased is respected. However, the level of authority given to families in each jurisdiction (de iure) can be ambiguous, and is often inconsistent with the role they actually play in practice (de facto) (REFERENCES ADDED Delgado et al 2019; Morla et al.), thus precluding the use of this dimension in our model of governance, which is based on societal awareness of actual policies.” 

 

Reviewer #3: 

The statistical analysis approach and software appear reasonable for this research. 

However there are concerns with the sample and the general presentation of the results.

1. The investigators claim to have a sample of 2006. Why is there a high percentage of females? 

This question converges with one observation made by Reviewer 1. Both reviewers are right to warn us of the possibility that our sample may be gender biased. We provide a twofold explanation for the significantly higher proportion of women than men in our sample. 

First, we have checked that in the Universities and in the study programs where we recruited the participants there is already a greater proportion of female students. We have also checked that these numbers mirror actual gender distribution in European tertiary studies, as reported by Eurostat (the reference is included in the text). Second, we suggest that the still high percentage of female students in our sample can be attributed to an answer bias that has also been detected in other studies (we provide two references where a similar effect has been identified). 

To give more prominence to this potential limitation in our study, we have introduced changes in the Methods section (p12), and in the Discussion (p25). 

(Page 12)

→ “Overall, 2006 Austrian, Belgian, Danish, German, Greek, Slovenian and Spanish students (age bracket (mode) = 20 to 24 years) took part in our study. A majority of them (74%) were women (see Table 1), which partially reflects an overrepresentation of women in the disciplines concerned [Supplementary file] (REFERENCE ADDED Eurostat), but may also account for an answer bias, as mentioned in the Discussion section”.

(Page 25)

→ “Second, our sample shows a gender bias across all countries and study fields. The fact that more women took part in the survey than did men can be partially explained by high female-to-male ratios in the sampling frame (supplementary file) –particularly among majors in health sciences (54-72%; vs 39-62% for humanities/social sciences)– as well as in the European student population (72% for health sciences students; 64% for humanities/arts/social sciences students; see [REFERENCE ADDED EUROSTAT, 2018]). Controlling for the gender distribution in the population, we still observe some overrepresentation of women in our sample, which may stem from sex differences in non-response bias, previously detected in other studies [19, REFERENCE ADDED Damman et al] indicating a higher retention and completion rates among women than men.” 

What is the general size of the population from which the sample is drawn and how representative is this sample of the population and in particular of the opt in and opt out countries as well as the disciplines being sampled? 

In the revised version of our manuscript (Abstract, Introduction, and Methods sections), we clarify that our study employed convenience sampling methods (see pages 1, 5 and 11). Neither our recruitment methods, nor the demographic sections of our study, were designed in order to ascertain the representativeness of the samples. As such, we now de-emphasize the precise values we obtained for each country, and focus on evaluating the strengths and limitations of our proposed model. 

Additionally, throughout the paper, we now emphasize the non-representative nature of our study samples (Introduction, page 6). 

(Pages 5-6)

→ “Our sample (N = 2006) serves the purpose of testing the theoretical framework, but the respondent population is highly selective and cannot be considered representative of national populations, as discussed below”. 

In the Limitations section (p.27) we also acknowledge the importance of generalizing these findings to nationally representative samples throughout Europe in order to better establish the governance quality across several countries, considering the general population as a whole.

(Page 26)

→ “Future work should use nationally representative population data to conduct the same governance analysis”. 

What statistical plan was in place to determine that 2006 was, in fact, the sample needed for this survey?

We established a target sample size per country of 200 participants –which we met and exceeded at every site except for Slovenia and Greece. According to sensitivity power analyses, this target sample size provided sufficient statistical power to detect small effects in our main analyses of interest (i.e., knowledge and attitudes; see Methods section, p14).

We set the 𝛼 level to .05 and power (1 - 𝛽) to .80 and conducted power analyses for one-sample proportion (knowledge rates) and t-tests (support/opposition attitudes). A target sample size of 200 participants per country (mean n per country = 287) provided sufficient statistical power to reliably detect small effects (Cohen’s g = .10, Cohen’s d = 0.20) in each separate country. This information has been included as a new paragraph in the Methods section, page 14. 

(Pages 13-14)

→ “To evaluate whether our sample size afforded adequate statistical power to examine variation in knowledge and attitudes across countries, we conducted sensitivity power analyses for one-sample proportion tests (i.e., knowledge) and t-tests (i.e., attitudes). Setting the alpha-level to .05 and power (1 - beta) to .80, a per-country target sample size of 200 (mean n/country = 287) enabled our study to reliably detect small effects on knowledge (Cohen’s g = .10) and attitudes (Cohen’s d = 0.20) in each country. In turn, our aggregate sample size (N = 2006) provided almost perfect power (> 99%; alpha = .05) to detect small effects even in multivariate analyses (i.e., f2 = .02).” 

2. What bias checks were made of the sample to assume its validity in this context?

The validity of the sample and methods was checked as follows:

With regard to the instrument’s content validity, we conducted several peer review rounds prior to administering the survey, first among the researchers involved, then seeking help from external colleagues in different countries. We then translated the questionnaire from English to each countries’ language, and then we back translated it to English to ensure accuracy. Finally, we used the resulting questionnaire to conduct pilot tests with lay people in two countries –Germany and Spain–, followed by interviews with the respondents.

With regard to the validity of the sample (a population integrated by university students overrepresented by women who was surveyed about public policies on organ donation), we have analyzed whether students’ knowledge and attitudes typically differ from the rest of the population with regard to organ donation, and whether women typically differ from men with regard to this topic.

To address these two questions, we have analyzed results from previous surveys on knowledge and attitudes toward organ donation policies. Two systematic reviews identified, respectively, 13 studies before January 2008 (Rithalia et al. 2009) and 60 studies from 2008 to 2018 (Molina-Pérez et al. 2019). In addition, we also checked the European Commission’s Special Eurobarometers n° 272 on organ donation (2007), n° 333a on organ donation and transplantation (2010), and n° 426 on blood, cell, and tissue donation (2015).

The Special Eurobarometers (SE) show no significant difference by age or gender among Europeans on awareness of organ donation laws (SE 2010), willingness to donate their own organs after death or organs from a deceased close family member (SE 2007, 2010), and no difference by age on willingness to donate tissues after death (SE 2015). However, all three studies indicate that education level was a socio-demographic discriminator: people educated till age 20 or later were more likely to be aware of the law and to support organ or tissue donation after death. This is relevant for our governance quality assessment model, which takes policy awareness as one of its main variables. The new version of the manuscript acknowledges this in the Limitations section (p. 26, see below).

The Eurobarometers’ results are consistent with nationally representative studies conducted in Germany, Spain, England, Denmark, Slovenia and Greece. In Germany, Decker and colleagues (2008) found that men and women did not differ regarding possession of an organ donor card. While women were more generally willing to donate organs after death than men, the difference was less than 3% (women 62.1% v. men 59.2%). No significant gender differences were found in the attitudes toward the different forms of consent legislation. In Spain, Conesa and colleagues (2003) found no gender effect on attitudes toward organ donation but some effect of age and education, with younger (<35 yo) and higher educated respondents (high school, university) being more favorable than older (>50 yo) and lesser educated respondents. In a more recent Spanish study, Scandroglio and colleagues (2011) found that “the disposition towards donation varies as a function of the level of social insertion, reflecting a common result in the literature, in which reticence is associated with a lower socio-economic or cultural level and more advanced ages”. In England, Webb and colleagues (2015) did not find significant differences by age or gender on overall willingness to donate, although there were differences when considering specifics (willingness to donate all organs or only some of them). In Denmark, Nordfalk and colleagues (2016) found that younger people were most strongly in favor of organ donation and also strongly opposed to a presumed consent system, but this study did not specify how much different the students’ attitudes were in relation to the average willingness to donate (85%). In Slovenia, Berzelak and colleagues (2019) reported that the donation willingness tends to increase with education (although the effect only reaches significance when comparing the lowest and highest educated respondents) and was significantly higher among women than men, with no significant effect of age (except for older respondents). In Greece, studies on students (Symvoulakis et al. 2014; Katsary et al. 2015) report higher levels of law awareness than a study on patients attending Greek general practices in a rural and urban setting (Symvoulakis et al. 2013). However, the Greek law changed from opt-in to opt-out between the patients’ study and the students’ studies, making it difficult to assess how much of the difference in their responses can be attributed to the sampling.

In general, it is difficult to assess how much students' responses in a given country vary from the general population because, as pointed out by our previous systematic review (Molina-Pérez et al. 2019), most general public surveys have been conducted in opt-in countries (and most are not peer reviewed scientific papers but grey literature funded by governments or national transplant organizations), while most surveys on students have been conducted in opt-out countries.

Conversely, there are some indications that students’ knowledge and attitudes towards organ donation, while different, are not at odds with the general population. In Germany, we conducted a students survey at the University of Göttingen in 2008/09 and again in 2014/15 (Schicktanz et al. 2017) whose results are consistent with the only available representative German survey conducted by the Federal Centre for Health Education (BzgA) also in 2008 and 2014. In Spain, a study on medical students—carried out in 2010-11—reported a willingness to donate of 80% (Ríos et al. 2019), which is approximately 15% above the level reported in the general public in Spain (Scandroglio et al. 2011; Conesa et al. 2005). More recently, we conducted a representative study in Spain's Andalusia region that shows no significant effect or only limited effect of age, education, and gender on knowledge and attitudes toward organ donation policies (Díaz-Cobacho et al., submitted for publication).

In sum, based on the analysis of these studies, it can be inferred that, overall, women’s knowledge and attitudes do not typically differ from men with regard to organ donation, and that students have higher awareness of organ donation policies than non students.

Moreover, since the aim of our study is to compare countries among each other, we believe that the presence of a sampling bias does not necessarily affect the validity of the study as long as this bias is present across countries. 

The new version of the manuscript accounts for these validity checks as follows:

(Page 24)

→ “This study has some limitations. First, the surveyed sample was drawn from the university student population, which is overrepresented by women, and highly non-representative of the general population of key variables such as age, educational attainment, and socioeconomic status. Reassuringly, however, previous studies on knowledge and attitudes toward organ donation policies in Europe show no significant effects of age or gender on awareness of organ donation laws (REFERENCE ADDED Special Eurobarometer 2010), willingness to donate their own organs after death or organs from a deceased close family member (REFERENCE ADDED Special Eurobarometer 2007, 2010), and no age differences in willingness to donate tissues after death (Special Eurobarometer 2015). Still, education level has been found to predict donation-related attitudes: enrollment in tertiary studies is associated with greater awareness of the law and support for posthumous organ or tissue donation. (REFERENCES ADDED Special Eurobarometer 2007, 2010, 2015). Therefore, we can expect that nationally representative studies would show lower levels of policy awareness, resulting in lower national scores in policy governance quality assessments. / “However, the education bias can be seen as less problematic for our approach as we were interested in the relation of attitudes and knowledge, and less on absolute knowledge levels. Since the aim of our study is to compare countries among each other, the presence of a sampling bias does not necessarily affect the validity of the study as long as this bias is present across countries.” 

Regarding sampling bias, we have clarified in the current version that our study was conducted on a convenience sample. In addition, we checked the gender bias in our sample. The higher presence of women aligns with the greater proportion of women in the sampling frame (i.e., the study programs from which we recruited our sample), and also in the university student population in Europe. The other potential bias that we have considered is the non-response bias. It is difficult to identify whether the reasons why some students decided not to participate in the study are relevant factors for the purpose of the study. 

The new version of the manuscript emphasizes these limitations as follows:

(Introduction, Page 5-6)

→ “Our sample (n=2006) serves the purpose of testing the theoretical framework, but the respondent population is highly selective and cannot be considered representative of national populations, as discussed below”. 

(Methods, Page 12)

→ “Overall, 2006 Austrian, Belgian, Danish, German, Greek, Slovenian and Spanish students (age bracket (mode) = 20 to 24 years) took part in our study. A majority of them (74%) were women (see Table 1), which partially reflects an overrepresentation of women in the disciplines concerned [Supplementary file] [REFERENCE ADDED EUROSTAT], but may also account for an answer bias, as mentioned in the Discussion section”.

(Pages 25)

 → “Second, our sample shows a gender bias across all countries and study fields. The fact that more women took part in the survey than did men can be partially explained by high female-to-male ratios in the sampling frame (supplementary file) –particularly among majors in health sciences (54-72%; vs 39-62% for humanities/social sciences)– as well as in the European student population (72% for health sciences students; 64% for humanities/arts/social sciences students; see [REFERENCE ADDED EUROSTAT, 2018]). Controlling for the gender distribution in the population, we still observe some overrepresentation of women in our sample, which may stem from sex differences in non-response bias, previously detected in other studies [19, Damman et al] indicating a higher retention and completion rates among women than men.” 

(Page 27) 

→Third, in comparison to other surveys about organ donation [20], the present questionnaire was longer and more time-consuming, so this can also select for highly motivated participants. This reason may account for some students’ decision not to participate in the study.

3. The investigators use p-values in general. However, they use terms or expressions such as ‘preponderance’ and ‘tendency to support’. What is the quantitative setting for these terms?

We describe statistical significance above the midpoint as ‘tendency to support’, and statistical significance below the midpoint as ‘tendency to oppose’. This applies to both the one-sample proportion tests, and the one-sample t-tests. We have clarified this on pages 15 and 17

(Page 14)

→ “Our first analysis step includes a test of participants’ attitudes toward a basic premise of the SHC concept, namely, the preference for public involvement in policy decisions. To assess this question, we conducted a series of one sample t-tests against the point of neutrality (μ = 3.5)”. 

(Pages. 16)

→ “we sought to understand whether countries differ in the extent to which participants favor the policy in place in their own country. For instance, support (/opposition) among German students would be defined by their favorable (/unfavorable) attitudes toward the opt-in system, whereas support (/opposition) among Spanish students would be defined by their favorable (/unfavorable) attitudes toward the opt-out system. We then conducted a series of one-sample t-tests against the point of neutrality (𝜇 = 3.5),. interpreting significant differences between national means and the scale midpoint as either a tendency to support the policy in place (if the national mean exceeds the midpoint), or a tendency to oppose the policy in place (if the national mean falls below the midpoint).” 

 

Supporting information file (ADDED INFORMATION ON SAMPLING FRAME)

REFERENCES

Berzelak N, Avsec D, Kamin T. Reluctance and Willingness for Organ Donation After Death Among the Slovene General Population. Slovenian Journal of Public Health. 2019;58: 155–163. doi:10.2478/sjph-2019-0020

Conesa C, Ríos A, Ramírez P, Canteras M, Rodríguez MM, Parrilla P. [Multivariate study of the psychosocial factors affecting public attitude towards organ donation]. Nefrologia. 2005;25: 684–697.

Decker O, Winter M, Brähler E, Beutel M. Between commodification and altruism: gender imbalance and attitudes towards organ donation. A representative survey of the German community. Journal of Gender Studies. 2008;17: 251–255. doi:10.1080/09589230802204290

Damman, O. C., Bogaerts, N. M. M., van den Haak, M. J., & Timmermans, D. R. M. (2017). How lay people understand and make sense of personalized disease risk information. Health Expect. 20, 973–983. doi: 10.1111/hex.12538

Delgado J, Molina-Pérez A, Shaw D, Rodríguez-Arias D. The Role of the Family in Deceased Organ Procurement. A Guide for Clinicians and Policy Makers. Transplantation. 2019;103: e112–e118. doi:10.1097/TP.0000000000002622

Katsari V, Domeyer PJ, Sarafis P, Souliotis K. Giving Your Last Gift: A Study of the Knowledge, Attitude and Information of Greek Students Regarding Organ Donation. Ann Transplant. 2015;20: 373–380. doi:10.12659/AOT.894510

Molina-Pérez A, Rodríguez-Arias D, Delgado-Rodríguez J, Morgan M, Frunza M, Randhawa G, et al. Public knowledge and attitudes towards consent policies for organ donation in Europe. A systematic review. Transplant Rev. 2019;33: 1–8. doi:10.1016/j.trre.2018.09.001

Morla M, Moya C, Delgado J and Molina A. European and comparative law study regarding family’s legal role in deceased organ procurement. Revista General de Derecho Público Comparado (accepted)

Nordfalk F, Olejaz M, Jensen AMB, Skovgaard LL, Hoeyer K. From motivation to acceptability: a survey of public attitudes towards organ donation in Denmark. Transplantation Research. 2016;5: 5. doi:10.1186/s13737-016-0035-2

Ríos A, López-Navas A, López-López A, Gómez FJ, Iriarte J, Herruzo R, et al. A Multicentre and stratified study of the attitude of medical students towards organ donation in Spain. Ethn Health. 2019;24: 443–461. doi:10.1080/13557858.2017.1346183

Rithalia A, McDaid C, Suekarran S, Norman G, Myers L, Sowden A. A systematic review of presumed consent systems for deceased organ donation. Health Technol Assess. 2009;13: iii, ix–xi, 1–95. doi:10.3310/hta13260

Rosenblum AM, Li AH-T, Roels L, Stewart B, Prakash V, Beitel J, et al. Worldwide variability in deceased organ donation registries. Transpl Int. 2012;25: 801–811. doi:10.1111/j.1432-2277.2012.01472.x

Scandroglio B, Domínguez-Gil B, Lopez J soler, Valentín MO, Martín MJ, Coll E, et al. Analysis of the attitudes and motivations of the Spanish population towards organ donation after death. Transplant international : official journal of the European Society for Organ Transplantation. 2011;24: 158–166. doi:10.1111/j.1432-2277.2010.01174.x

Schicktanz S, Pfaller L, Hansen SL, Boos M. Attitudes towards brain death and conceptions of the body in relation to willingness or reluctance to donate: results of a student survey before and after the German transplantation scandals and legal changes. J Public Health. 2017;25: 249–256. doi:10.1007/s10389-017-0786-3

Symvoulakis EK, Rachiotis G, Papagiannis D, Markaki A, Dimitroglou Y, Morgan M, et al. Organ donation knowledge and attitudes among health science students in Greece: emerging interprofessional needs. Int J Med Sci. 2014;11: 634–640. doi:10.7150/ijms.8686

Symvoulakis EK, Markaki A, Galanakis C, Klinis S, Morgan M, Jones R. Shifting towards an opt-out system in Greece: a general practice based pilot study. Int J Med Sci. 2013;10: 1547–1551. doi:10.7150/ijms.7027

Touraine J-L. Mission “flash” relative aux conditions de prélèvement d’organes et du refus de tels prélèvements. Paris: Assemblée Nationale; 2017 Dec. Available: https://www2.assemblee-nationale.fr/static/15/commissions/CAffSoc/Mission_flash_don_organes_communication_rapporteur_20171220.pdf

Webb G, Phillips N, Reddiford S, Neuberger J. Factors Affecting the Decision to Grant Consent for Organ Donation: A Survey of Adults in England. Transplantation. 2015;99: 1396–1402. doi:10.1097/TP.0000000000000504

WHO Guiding Principles on Human Cell, Tissue and Organ Transplantation. Cell Tissue Bank. 2010;11: 413–419. doi:10.1007/s10561-010-9226-0

---

## [Decision Letter · Decision Letter 1]

20 May 2021

Governance Quality Indicators for National Organ Procurement Policies. A novel approach based on a Cross-European Survey of Students’ Knowledge and Views about Consent Policies

PONE-D-21-01301R1

Dear Dr. Rodriguez-Arias,

We’re pleased to inform you that your manuscript has been judged scientifically suitable for publication and will be formally accepted for publication once it meets all outstanding technical requirements.

Kind regards,

Frank JMF Dor, M.D., Ph.D., FEBS, FRCS

Academic Editor

PLOS ONE

Additional Editor Comments (optional):

Reviewers' comments:

Reviewer's Responses to Questions

**Comments to the Author**

1. If the authors have adequately addressed your comments raised in a previous round of review and you feel that this manuscript is now acceptable for publication, you may indicate that here to bypass the “Comments to the Author” section, enter your conflict of interest statement in the “Confidential to Editor” section, and submit your "Accept" recommendation.

Reviewer #1: All comments have been addressed

Reviewer #2: All comments have been addressed

2. Is the manuscript technically sound, and do the data support the conclusions?

Reviewer #1: Yes

Reviewer #2: Yes

3. Has the statistical analysis been performed appropriately and rigorously? 

Reviewer #1: I Don't Know

Reviewer #2: Yes

4. Have the authors made all data underlying the findings in their manuscript fully available?

Reviewer #1: Yes

Reviewer #2: Yes

5. Is the manuscript presented in an intelligible fashion and written in standard English?

Reviewer #1: Yes

Reviewer #2: Yes

6. Review Comments to the Author

Reviewer #1: This is a well written article which deserves publication. My concerns have all been addressed and the responses are satisfactory

Reviewer #2: The revision of the manuscript has been performed thoroughly. All my questions have been answered and the information is added to the paper.

7. PLOS authors have the option to publish the peer review history of their article (what does this mean?). If published, this will include your full peer review and any attached files.

Reviewer #1: No

Reviewer #2: No

---

## [Editor Report · Acceptance letter]

28 May 2021

PONE-D-21-01301R1 

Governance Quality Indicators for Organ Procurement Policies 

Dear Dr. Rodríguez-Arias:

I'm pleased to inform you that your manuscript has been deemed suitable for publication in PLOS ONE. Congratulations! Your manuscript is now with our production department. 

Kind regards, 

on behalf of

Dr. Frank JMF Dor 

Academic Editor

PLOS ONE